# Deep neural networks and humans both benefit from compositional language structure

Lukas Galke [1,2] ✉, Yoav Ram[3,4] & Limor Raviv [2,5]

Deep neural networks drive the success of natural language processing. A fundamental property of language is its compositional structure, allowing humans to systematically produce forms for new meanings. For humans, languages with more compositional and transparent structures are typically easier to learn than those with opaque and irregular structures. However, this learnability advantage has not yet been shown for deep neural networks, limiting their use as models for human language learning. Here, we directly test how neural networks compare to humans in learning and generalizing different languages that vary in their degree of compositional structure. We evaluate the memorization and generalization capabilities of a large language model and recurrent neural networks, and show that both deep neural networks exhibit a learnability advantage for more structured linguistic input: neural networks exposed to more compositional languages show more systematic generalization, greater agreement between different agents, and greater similarity to human learners.

Compositionality, i.e., whether the meaning of a compound expression can be derived solely from the meaning of its constituent parts, has been studied for decades by both computer scientists and linguists[1–5]. In particular, languages differ in how they map meanings into morphosyntactic structures[6,7] and cross-linguistic studies find substantial differences in the degree of structural complexity across languages[8–14]. These differences can stem from multiple and often confounded aspects of linguistic structure including the degree of compositionality[7], which can be quantified by correlating differences in meaning with differences in form[15]. For example, the English term "white horse" is compositional since its meaning can be directly inferred given knowledge about its constituents "white" and "horse". In contrast, consider the equivalent German term "Schimmel", whose meaning cannot be derived from "weiß" (white) and "Pferd" (horse). Crucially, compositionality directly affects our ability to make systematic generalizations in a given language and thus shapes its immense expressive power—which also explains its high relevance in machine learning[1,2,16–24].

Importantly, cross-linguistic differences in compositional structure were suggested to impact human language learning and generalization in the real world[25–27] as well as in lab experiments[28–32], with more compositional linguistic structures typically being easier to learn for adult learners. In a large-scale artificial language learning study with adult human participants, the acquisition of a broad yet tightly controlled range of comparable languages with different degrees of compositional structure was tested[28]. Results showed that more compositional languages were learned faster, better, and more consistently by the adult learners, and that learning more structured languages also promoted better generalizations and more robust convergence on labels for new, unfamiliar meanings. This is likely because more systematic and compositional linguistic input allow learners to derive a set of generative rules rather than rote memorizing individual forms, and then enables learners to use these rules to produce an infinite number of utterances after exposure to just a finite set[32–36]. This learnability and generalization advantage for more structured linguistic input has far-reaching implications for broader

[1]Department of Mathematics and Computer Science, University of Southern Denmark, Odense, Denmark. [2]LEADS group, Max Planck Institute for Psycholinguistics, Nijmegen, Netherlands. [3]School of Zoology, Faculty of Life Sciences, Tel Aviv University, Tel Aviv, Israel. [4]Sagol School of Neuroscience, Tel Aviv University, Tel Aviv, Israel. [5]cSCAN, University of Glasgow, Glasgow, UK. ✉e-mail: galke@imada.sdu.dk

theories on language evolution in our species (and potentially other learning systems): A large body of computational models and experimental work with human participants show that more systematic and compositional structures emerge during cross-generational transmission and communication precisely because such structures are learned better, while still allowing for high expressivity[30,32–35,37–40]. Hence, popular theories of language evolution attribute the emergence of systematic and compositional structure in natural languages to such learnability pressures[32,41], suggesting a causal role not only in language learning, but also in shaping the way human languages are structured. To what extent this advantage of linguistic structure carries over to artificial learning systems is currently poorly understood—which is the aim of the current study.

Despite an increasing body of work that reports striking similarities between humans and large language models[42–48], and despite large language models being incredibly proficient at using language and generalizing to new tasks with little to no new training data[49–52], research on emergent communication suggests that deep neural networks (the class of models that underlies large language models) show no correlation between the degree of compositional structure in the emergent language and the generalization capabilities of the networks. In other words, unlike humans, artificial neural networks do not seem to benefit from more compositional structure when they are made to develop their own communication protocol, at least without dedicated intervention[53–56] (but see ref. [57]). Thus, this finding raises the question of whether systematic and compositional linguistic structure is helpful at all for deep neural networks, and to what extent compositionality affects the memorization and generalization abilities of deep neural networks learning a new language.

The mismatch with humans can potentially be explained by differences in model design and experimental procedure[58]. For instance, deep neural networks typically have immense model capacity due to overparametrization[59–64], which means they could easily memorize all individual forms without the need to identify compositional patterns[23,58]. A competing hypothesis is that neural networks do benefit from compositional structure in the data given that this structure is reflected in the statistical patterns of the data which impacts the optimization of the model parameters[65,66]. Specifically, in a language with a higher degree of compositionality, the individual units of meaning are reused in different contexts and thus appear more often in the training data, such that these recurring units of meaning and their contextualization patterns are learned better because of the repeated presentation throughout training (see refs. [24,67]).

Here, we explore this precise relationship between compositional structure and generalization with deep neural networks. The central question we aim to answer is: Do deep neural network models exhibit the same learning and generalization advantage when trained on more structured linguistic input as human adults? Specifically, we investigate whether the advantage of compositionality in language learning and language use carries over to artificial learning systems, while considering GPT-3.5 as a pre-trained large language model and a custom model architecture based on recurrent neural networks (RNNs) trained from scratch. Our work contributes to the understanding of deep neural networks and large language models, sheds new light on the similarity between humans and machines, and, consequently, opens up future directions of simulating the very emergence of language and linguistic structure with deep neural network agents.

To allow for direct comparisons between humans and machines, we carefully follow the experimental procedure and measures of a recent large-scale preregistered language learning study with adult participants[28]. We consider 10 input languages, each of which has emerged independently and spontaneously through a group communication experiment with adult human participants[68]. The languages describe four different novel shapes moving on the screen in a different direction (0-360 degree), and vary in their degree of compositional structure: ranging from fully idiosyncratic languages with entirely different labels for two related meaning (e.g., 'kuim' and 'goom' for the same shape moving into a different direction) to highly structured languages, which re-use parts of the descriptive label (e.g., referring to the two scenes as 'fest-ii' and 'fest-ui'). See Fig. 1. Neural networks were then trained on the exact same stimuli presented to humans and in the same order, using the same learning tasks, providing the same feedback during learning blocks, and evaluated with the same memorization and generalization tests. Figure 1 shows the recurrent neural network architecture and summarizes the experimental procedure: Full details of the experimental setup, custom recurrent neural neural network models, and how we employed large language models are provided in the Methods section.

By evaluating the performance of small and large language models across languages with varying degrees of compositional structure, we show that more structured linguistic input results in more systematic generalization, greater agreement between different agents, and closer alignment with human learning patterns. Our findings show that neural networks trained on highly structured languages produce more transparent generalizations, with memorization and generalization patterns that become increasingly human-like as linguistic structure becomes more compositional. Implications of our work extend to the design of artificial agents and the understanding of human language learning, suggesting that the systematicity of linguistic input plays a crucial role in shaping the learning dynamics of both artificial and natural language systems.

## Results

To preview our results, we find a consistent advantage of more systematic and compositional linguistic structure for learning and generalization, closely reflecting adult human participants. The generalization behavior of both large language models (pre-trained on other languages) and recurrent neural networks (trained from scratch) was far more systematic and transparent when the input languages were more compositional. Moreover, recurrent neural network agents displayed a higher agreement with other agents as well as with humans when the input was more compositional, leading to converging transparent generalizations for new unseen input. A glossary of evaluation metrics can be found in Table 1. More detailed descriptions of the metrics are provided in the Methods section.

### More compositional structure leads to higher similarity to humans and more systematic generalization of large language models

We first test whether large language models benefit from compositional structure when learning a new language. Such language models are pre-trained to predict left-out words in web-scale corpora of text data, leaving them with high competence in at least one language, similar to adult human participants. Specifically, we employ the large language model GPT-3.5 (version text-davinci-003) which is capable of in-context learning, i. e., having the model tackle a new task only based on a few examples in the prompt[49,69]. We make use of this property to evaluate the model in learning the new languages. For each input language, we insert the form-meaning pairs in the prompt of the large language model, followed by a single meaning for which the label needs to be completed. We repeat this procedure multiple times to have the language model produce labels for the memorization test (known meanings) as well as the generalization test (new meanings).

In the generalization test, there is no true label in the input language. To capture the degree to which new labels conform to the labels of the input language (i.e., to what extent the generalization is systematic), we correlate the pairwise label difference and the pairwise semantic difference between the labels generated for new scenes and the labels generated by the same agent for known scenes[28].

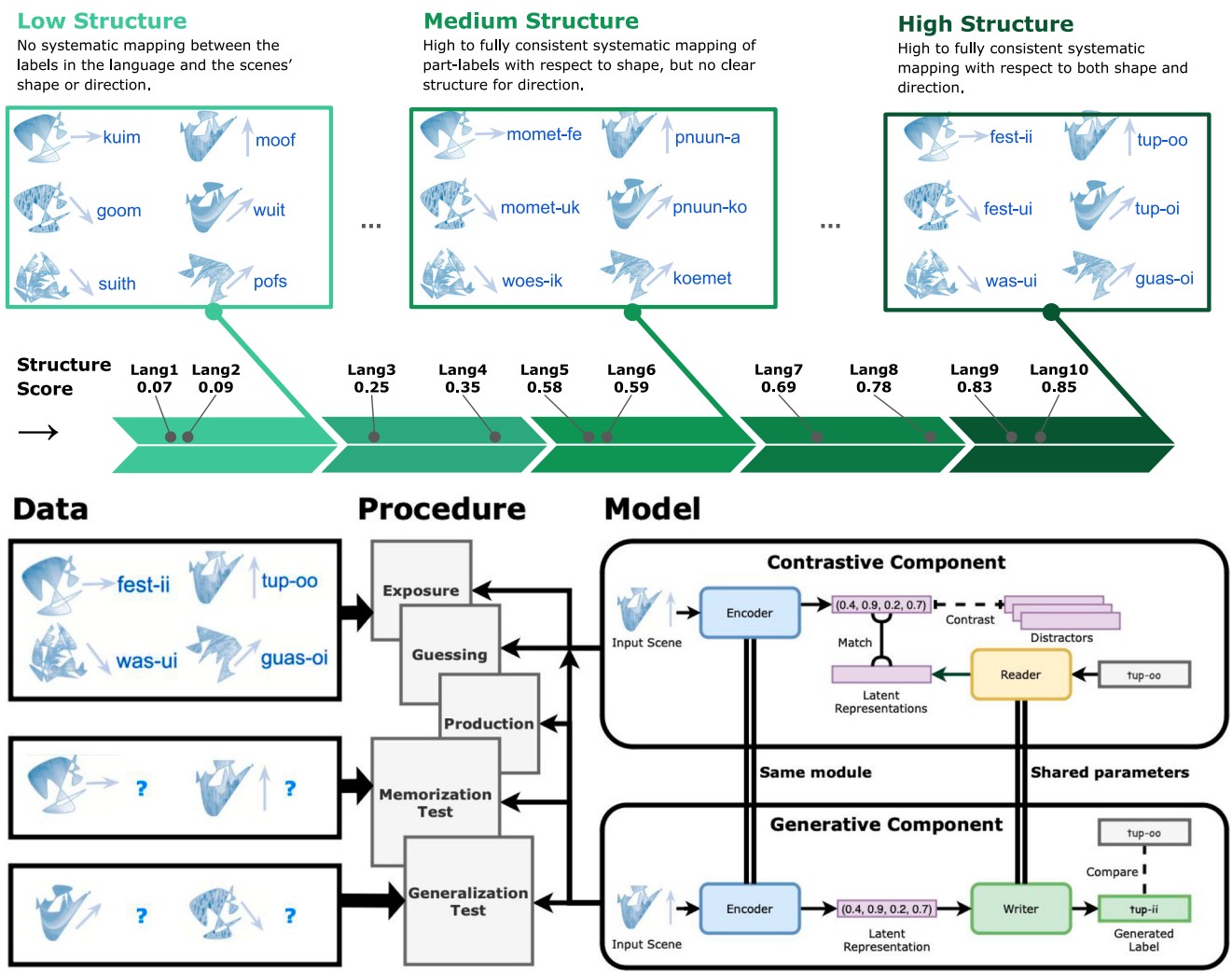

**Fig. 1 | Overview.** Overview of input languages (Top), the experimental procedure (Bottom Center) along with exemplary input data from one language (Bottom Left), and the model architecture (Bottom Right). Low-structured input languages show no signs of systematicity or compositionality, whereas high-structured languages are systematic and compositional with respect to both attributes: shape and angle. For each language, we train the model for multiple rounds of exposure, guessing, production. After each round, we conduct a memorization test to evaluate productions for previously seen items and a generalization test evaluating the productions for new items. Graphical elements in the upper part of this figure are re-used and adapted with permission from Raviv et al.[68].

Strikingly, the results reveal that a higher degree of compositional structure in the input language leads to generalizations that are more systematic (see Fig. 2B), closely reflecting the pattern of adult human learners (Fig. 2A). Table 2 shows examples of the final productions of humans and large language models during generalization (more examples are provided in Supplementary Tables 4 and 5).

In addition, we evaluate the production similarity as character-level length-normalized edit distance between the generated labels and labels produced by human participants during generalization. The results show that, given more structured linguistic input, GPT-3.5 also yields productions that are more similar to the productions of human participants, calculated as the average similarity between GPT-3.5's production and all human productions for the same scene in the same language (Fig. 3B). Analogously, Fig. 3A shows the similarity of humans to other human learners during generalization.

We then conduct an error analysis to understand better whether the memorization errors are similarly affected by the degree of compositional structure. We analyze the cases where the learning system fails to memorize the correct label perfectly and calculate the production similarity (1 minus length-normalized edit distance). Again, the results show the same pattern for adult human participants and large language models (see Fig. 4A, B): When there is more structure in the input language, the non-perfectly memorized productions are more similar to the correct labels.

### More compositional structure leads to higher similarity to humans and more systematic generalization with recurrent neural networks

In addition to large language models, we test a custom neural network architecture trained from random initialization, which allows us to conduct a close analysis of the learning trajectory. Our custom model architecture is designed to simulate the exposure, guessing, and production blocks that human participants have engaged in (see Fig. 1). The architecture is inspired by image-captioning approaches[70], the emergent communication literature[71], and in particular, our recent review paper[58] which suggested having shared model parameters between generation and processing of a label. Our model consists of two components: a generative component that facilitates the production of a descriptive sequence of symbols (here, a label) for a scene, while a contrastive component shapes the latent space and enables the models to carry out guessing tasks during learning (i.e., given a label, pick the correct scene from a set of distractors). Each component has a

**Table 1 | Glossary of Metrics**

| Metric | Description |
|---|---|
| Production Similarity | One minus length-normalized edit distance |
| Semantic Difference | Sum of the difference in shape (1 if different and 0 otherwise) and the absolute difference in angles (divided by 180) |
| Structure Score | Pearson correlation between (a) pairwise semantic differences and (b) pairwise length-normalized edit distances, where (a) and (b) are calculated on all pairs of items in the original input language |
| Generalization Score | Pearson correlation between (a) pairwise semantic differences and (b) pairwise length-normalized edit distances, where (a) and (b) are calculated on all pairs between productions for memorized items and productions for generalized items |
| Convergence Score | Average of all values for item-level production similarity for the same items between different learners trained on the same language |
| Human Label Similarity | Item-level production similarity to (other) human learners, averaged across different human learners |
| True Label Similarity | Item-level production similarity to input language |

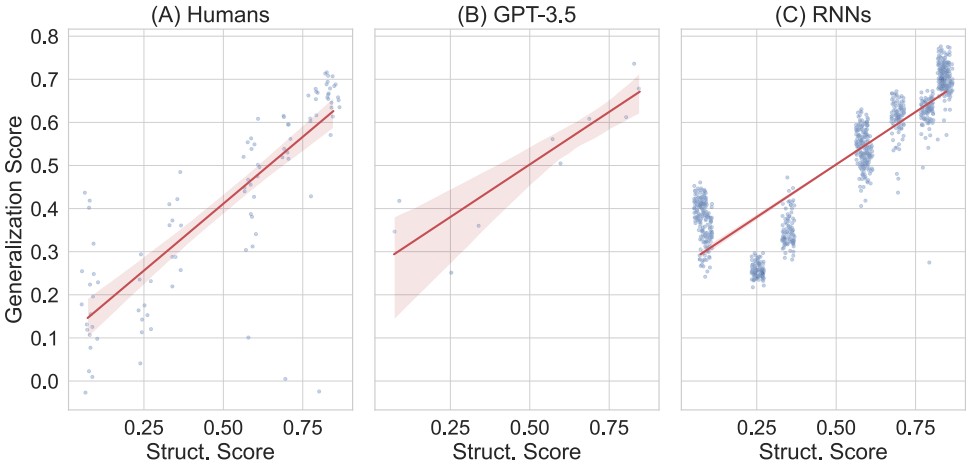

**Fig. 2 | More structure in the input language leads to more systematic generalization for all three learning systems.** Final generalization score achieved by humans (**A**), GPT-3.5 (**B**), and recurrent neural networks (**C**) for each of the input languages. The x-axis shows the structure score of the input languages. Each point corresponds to the generalization score calculated for the entire input language. This score reflects the degree to which learners systematically generalized new labels relative to the labels they learned. For example, generalization score would be high if learners successfully recombines previously used parts, e.g., combining 'muif' for the shape and 'i' for the direction into 'muif-i'. Error regions of the regression lines are 95% confidence intervals estimated via bootstrapping.

sequential recurrent neural network module to carry out the generation and processing of a sequence, respectively, for which we use the well-known long short-term memory[72]. The symbol embedding that maps each symbol of the sequence into a continuous vector is shared between the generative and the contrastive component. Moreover, the two components share the same encoder module that transforms an input scene into a latent representation, which then serve as the initial state of the generative component. Production tasks are modeled by a generative objective: Based on this initial state, the model generates a label, character by character. This generated label is compared to the target input language by character-level cross-entropy. Guessing tasks are modeled by a contrastive objective[73], which aligns the latent representation of input scenes and corresponding labels and facilitates selecting the correct scene from a set of distractors. As the encoder is shared, the contrastive objective shapes the space of initial states of the production model.

In total, we trained 1000 neural network agents with different random seeds (100 for each of the ten input languages) and calculated the following measures after each training round: the similarity between networks' productions and the input language; the similarity between networks' productions and the human learners' productions during memorization and generalization; the generalization score capturing the degree of systematicity; and a convergence score capturing the agreement between different agents. We evaluated these measures after each of 100 rounds.

The results are shown in Fig. 5. Extended results can be found in Supplementary Figs. 1–6. In the following, we present the results for the learning trajectory organized along the two types of tests: memorization and generalization, before presenting the final results of RNNs.

**Memorization trajectory.** How well did neural agents memorize the input languages? And how similar were their generated labels to those produced by human learners during generalization? This is measured by production similarity[28], which captures the similarity between the original label and the produced label by calculating the average normalized edit distance between two labels for the same scene. We use this measure in two ways: once to compare the generated labels to the true label of the input language and once to compare the machine-generated label to the human-generated label for the same scene.

**Similarity to input languages during memorization.** With sufficient training rounds, all languages can be learned by all neural network agents, reaching a production similarity of at least 0.8 (out of 1) by round 60 (Fig. 5A). Structured languages are learned significantly better (LME 1; $\beta = 0.045$, SE = 0.001, $z = 62.865$, $p < 0.001$), i. e., they show a higher similarity with the input language. However, this advantage tends to diminish over training rounds (LME 1; $\beta = -0.005$, SE < 0.001, $z = -54.978$, $p < 0.001$).

**Similarity to humans during memorization.** We measure the similarity to humans during memorization (i.e., comparing productions of both learning systems after completing the training rounds) and the memorization test data of the neural network agents after each

training round (Fig. 5B). More compositional input languages led to a significantly greater similarity with human learners (LME 2; $\beta$ = 0.097, SE = 0.001, $z$ = 81.429, $p$ < 0.001). This effect became even stronger over rounds (LME 2; $\beta$ = 0.022, SE < 0.001, $z$ = 208.708, $p$ < 0.001).

**Generalization trajectory.** We evaluate the productions of neural agents when they generalize, i.e., produce labels for new scenes that were not part of the training data. We test the productions regarding three aspects: the degree of systematicity, the similarity to humans, and the generalization convergence between different agents. As with large language models, we evaluate the generalization score. More structured languages consistently led to significantly higher generalization scores (Fig. 5C) (LME 3; $\beta$ = 0.088, SE = 0.001, $z$ = 148.901, $p$ < 0.001), and this effect became stronger with time ($\beta$ = 0.046, SE < 0.001, $z$ = 703.483, $p$ < 0.001).

**Table 2 | Generalization examples from neural network and human learners, showing labels generated for unseen scenes**

| Struct. | Shape | Angle | Human | RNN | GPT-3.5 |
|---|---|---|---|---|---|
| low | 2 | 360 | kokoke | seefe | tik-tik |
| | 4 | 45 | woti | kite | hihi |
| | 3 | 150 | ptiu | mimi | hihi |
| mid-low | 3 | 225 | wangsuus | wangsoe | wangsuus |
| | 4 | 225 | gntsoe | gntuu | gntsii |
| | 1 | 135 | sketsi | gesh | geshts |
| mid | 3 | 60 | powi | powu-u-u | powee |
| | 4 | 330 | fuottoa | fuotio | fuottu-u-u |
| | 1 | 30 | fewo-o-o-o | fewen | fewee |
| mid-high | 1 | 30 | fas-a | fas-a | fas-a |
| | 3 | 360 | muif-i | muif-a | muif-i |
| | 1 | 225 | fas-huif | fas-huif | fas-huif |
| high | 4 | 60 | smut-tkk | smut-tk | smut-ttk |
| | 2 | 360 | nif-k | nif-kks | nif-k |
| | 1 | 315 | wef-ks | wef-kks | wef-kks |

The column GPT-3.5 corresponds to completions generated by the GPT-3.5 model text-davinci-003 via in-context learning, where the training data is provided in context. The examples cover the differently structured input languages from low to high.

**Similarity to humans during generalization.** We measure the similarity between the productions of neural network agents and humans for new scenes (Fig. 5D), i. e., during generalization. Examples are shown in Table 2. More structure in the input language led to a significantly higher similarity between humans and neural agents (LME 5; $\beta$ = 0.132, SE = 0.002, $z$ = 70.280, $p$ < 0.001), which became stronger over rounds ($\beta$ = 0.046, SE < 0.001, $z$ = 344.287, $p$ < 0.001).

**Convergence between neural agents during generalization.** More structured languages lead to better agreement between networks (LME 4; $\beta$ = 0.043, SE = 0.001, $z$ = 49.027, $p$ < 0.001), such that, for more structured languages, different neural agents learning the same input language produced more similar labels for new scenes (Fig. 5E). This effect became stronger over rounds ($\beta$ = 0.009, SE < 0.001, $z$ = 121.740, $p$ < 0.001).

**Final results of RNNs.** To compare our custom recurrent neural network agents with large language models and with humans, we visualize the relationship between compositional structure of the input language and final generalization performance in Fig. 2C. All three learning systems (Humans, RNNs, and GPT-3.5) show the same trend: more compositionality in the input language leads to more systematic generalization.

Moreover, we calculate the average similarity to generalizations of human participants on the same language and item. Comparing the productions during generalization, the results show that a higher degree of structure in the input language leads to more similarity with humans (see Fig. 3C). This pattern of compositional structure leading to more human-like generalizations is present in both RNNs' and GPT-3.5's generated labels—as well as when comparing humans to other humans (see Fig. 3).

Lastly, we visualize the results of the memorization error analysis for recurrent neural networks alongside humans and GPT-3.5 in Fig. 4. The pattern is the same for all three different learning systems, be it artificial or biological: more compositional structure leads to errors that are more similar to the true label.

## Discussion

Our results show that deep neural networks benefit from more structured linguistic input as humans do and that neural networks' performance becomes increasingly more human-like when trained on more

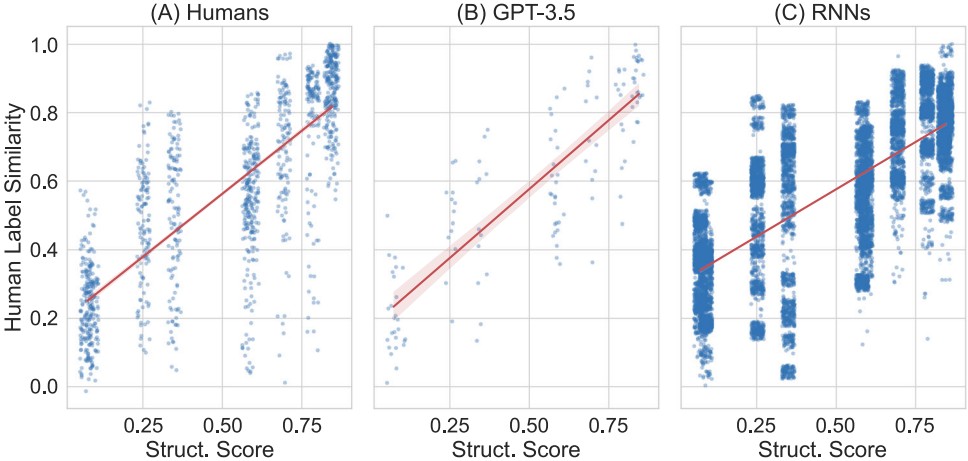

**Fig. 3 | More structure in the input language leads to more similarity to human participants for both RNNs and GPT-3.5.** Final similarity to humans during generalization: Final production similarity with (other) human participants during generalization achieved by humans (**A**), GPT-3.5 (**B**) and recurrent neural networks (**C**) for each of the input languages. The x-axis shows the structure score of the input languages. Each point corresponds to the production similarity score (calculated as length-normalized edit distance) between humans' productions and models' productions for every item in the language. For example, a recurrent neural network that produced 'muif-a' for shape 3 moving in direction 360 degrees would have a high production similarity to the majority of human participants who produced 'muif-i'. Error regions of the regression lines show 95% confidence intervals estimated via bootstrapping.

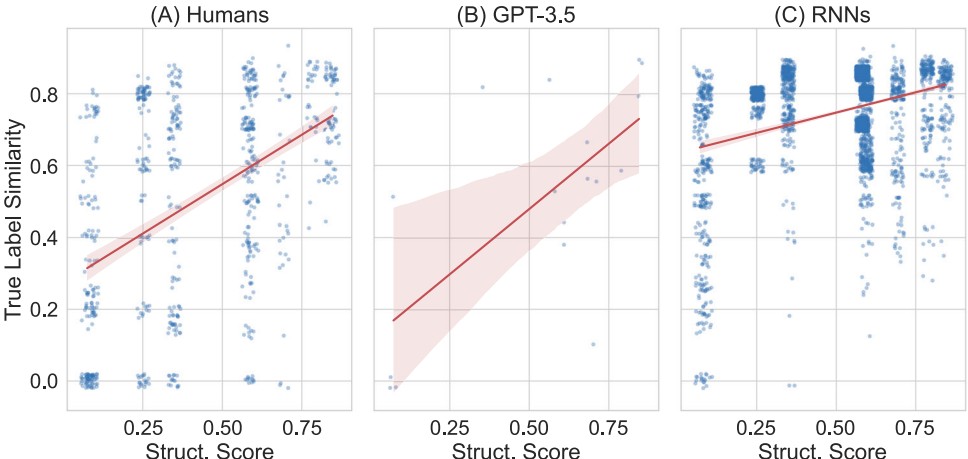

**Fig. 4 | More structure leads to erroneously memorized examples being more similar to the ground truth of the input language.** Memorization error analysis of human participants (**A**), GPT-3.5 (**B**), and recurrent neural networks (**C**). The error rates are 33.30% for humans, 7.39% for GPT-3.5 via in-context learning, and 13.87% for RNNs after 100 epochs of training. The x-axis shows the structure score of the input language. Each point corresponds to the production similarity score (calculated as length-normalized edit distance) between an erroneously memorized label for a given item and the correct corresponding label as it appears in the input language. For example, 'wangsus' has a higher similarity with 'wangsuus' than 'gempt'. Error bands of the regression lines show 95% confidence intervals estimated via bootstrapping.

structured languages. This structure bias can be found in networks' learning trajectories and even more so in the networks' generalization behavior, mimicking previous findings with humans. Although all languages can eventually be (almost) perfectly learned, we show that more structured languages are learned better and more similarly to human productions. Deep neural networks and humans produce nearly identical labels when trained on high-structured languages but not when trained on low-structured languages. Moreover, networks that learn more structured languages are significantly better at systematic generalization to new, unseen items, and crucially, their generalizations are significantly more consistent and more human-like. This means that highly systematic grammars allow for better generalization and facilitate greater alignment between different neural agents and between neural agents and humans. We have replicated these results with small recurrent neural networks and with transformer-based large language models, showing that, together with humans, all three learning systems show the same bias in systematic generalization and memorization errors. Thus, our findings strengthen the idea that language models are useful for studying human cognitive mechanisms, complementing the increasing evidence of similarity in language learning between humans and machines[42–48].

Specifically, we find very similar effects of structure on generalization and on the similarity to humans across all three learning systems. While we find a different slope for humans and RNNs in the memorization error analysis (likely due to RNNs being less impacted by memorization difficulty given sufficient training), the general trend is consistent: for both humans and artificial agents, exposure to more structured languages leads to production errors that are nevertheless more similar to the correct labels (i.e., their errors are less "wrong").

We assume that the reason for the increased similarity between machines and humans is that the ways to generalize are more transparent in high-structured languages, while there are none or less transparent generalization patterns available in low- and medium-structured languages. This leads both humans and neural networks to a higher production variation in lower structured languages, as different options on how to generalize are equally likely. This point is well supported by results from humans, who indeed show increased convergence between participants when learning higher structured languages[28] Our results thereby demonstrate that what is more transparent for humans is also more transparent for deep neural networks.

Analyzing the learning trajectory of recurrent neural networks, we find that languages with mid and mid-low structures often show an advantage in both memorization and generalization during the early stages of learning. This may be due to the fact that these mid-structured languages trade off full expressiveness with more simplicity (see Supplementary Table 1). For example, one of the mid-structured languages includes a marker for "moving on the diagonal", but does not distinguish the direction of the movement (e.g., center to north-east vs. center to south-west). As a result, the same label is used for two distinct meanings, which is easier to learn in the first place (less variation), but not sufficient to fully differentiate between items and thus harming systematic generalization.

As for implications, our findings first and foremost support the idea that languages' underlying grammatical structure can be learned directly from (grounded) linguistic input alone[35,41,74–76]. To ensure that the advantage of more structured linguistic input does not stem from the fact that the learning system was already proficient in a different language—i. e., as are pre-trained language models and adult humans—we also also considered models trained from random initialization. Therefore, our results predict that children would also benefit from more systematic compositional structure in the same way adults do—a prediction we are currently testing (preregistration:[77]).

Our findings have further implications for machine learning, where systematic generalization beyond the training distribution (out-of-domain) is of high interest[17,19–21,78]. Systematic in-domain generalization, as studied here, is a critical prerequisite for systematic out-of-domain generalization. Specifically, we show that seeding a learning system with well-structured inputs can improve their ability to systematically generalize to combinations that were not observed during training. Even though our study is based on artificial languages, our findings directly pertain to the natural language processing of real-world languages. To confirm this prediction, we re-analyzed data from Wu et al.[14], who used the Wug Test[79] to test language models' ability to predict different forms of unfamiliar words in a wide range of natural languages. Indeed, we find that the Wug Test accuracy negatively correlates with the degree of irregularity of the language (Spearman's $\rho = -0.96$, $p < 10^{-15}$; Kendall's $\tau = -0.86$, $p < 10^{-14}$). This strong negative correlation suggests that natural languages with fewer irregularities, i. e., more consistently structured natural languages, are indeed easier to learn for machines.

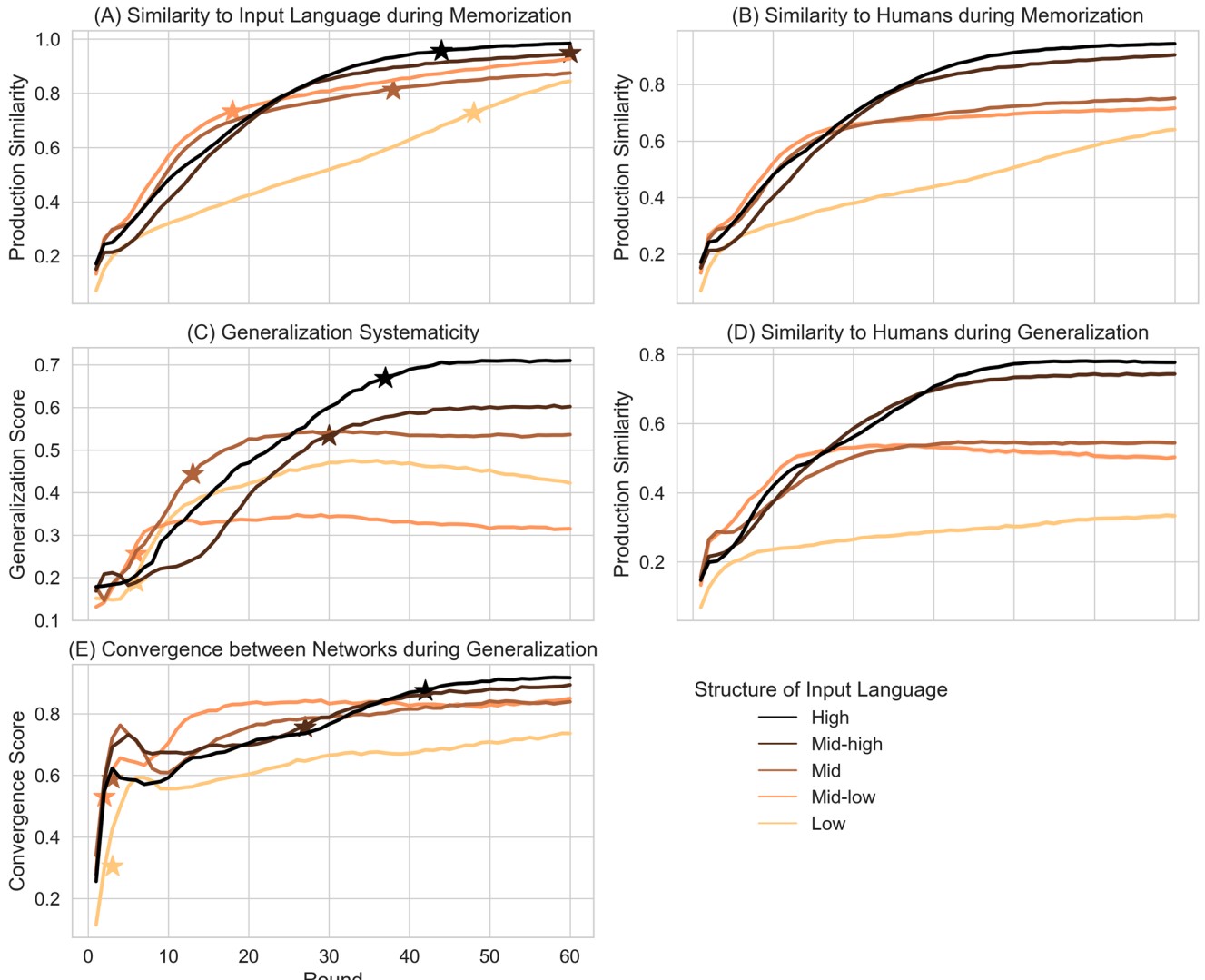

**Fig. 5 | Learning trajectory of recurrent neural networks' memorization and generalization performance.** More structured languages lead to better and faster reproduction of the input language (**A**), to better generalization on unknown scenes (**C**), better agreement with human participants during memorization (**B**) and generalization (**D**), and higher convergence between networks (**E**). **A** Production similarity between labels generated by neural agents and labels of the input language. (**B**): Production similarity between labels generated by neural agents and labels generated by human participants. **C** Generalization score of labels generated by neural agents for new scenes that were not part of the training data.

**D** Production similarity between labels generated by neural agents and labels generated by human participants for unseen scenes. **E** Convergence score measures the similarity between labels generated for unseen scenes by different neural agents. Stars mark the round at which neural agents first exceed the final performance of human participants. Input languages are grouped into 5 bins. Each line is the average of 200 neural agents with a different random initialization. A star marks the epoch at which the RNN agents exceed human performance. Results are cut off for visualization at epoch 60, full results in SI.

Crucially, there is a positive correlation between the degree of linguistic structure and population size[12,68,80,81], with low-resource languages (i. e., languages spoken by smaller communities for which there is only very little training data available) typically having less structured languages. Since our study predicts that such languages are harder to learn for deep neural networks, this results in a double whammy for developing natural language processing systems for small communities' languages—exacerbating challenges of low-resource language modeling[82]. Interestingly, the benefit of structured input could also explain the importance of highly-structured programming languages in the data mix for training large language models[83].

Finally, our results are of high relevance to the field of emergent communication. Emergent communication strives to simulate the evolution of language with multi-agent reinforcement learning[53,56,71,84–89]. However, as argued in the introduction, certain linguistic phenomena of natural language appear to be hard to replicate in multi-agent reinforcement learning[54,55,58,90], which had raised the question whether compositionality is helpful for neural networks at all. We hypothesized that these mismatches are caused by the lack of cognitive constraints[58] eradicating the learnability pressure underlying human language evolution[37]. Our findings support the importance of a learnability pressure for compositional languages to emerge. By confirming a result previously found in humans[28] in deep neural networks, we take the first steps to bring emergent communication closer to the field of language evolution, supporting simulations of language emergence with neural networks.

An interesting direction for future research is to investigate potential differences in the amount of training that a neural network needs compared to humans. Through anchoring our experiments in human data, we were able to directly identify the point during training at which recurrent neural networks equalize with human participants.

However, the location of this point depends on various factors such as the amount of data, the number of parameters that are optimized, and the number of optimization steps —which makes it challenging to predict this point in advance. While we have identified this point through analyzing the learning trajectory, our analysis does not depend on it, as all measures including the similarity between humans and machines are calculated based on productions taken at the end of training.

Furthermore, we have chosen to work with an input representation that we deemed easiest to process for each type of learning system. Since the particular way in which agents represent the visual world was not the object of the current study, our rationale here was to provide each learning system with a representation that is easiest or most natural to process. Human participants would have likely had a harder time finding patterns in attribute-value vectors, consisting of 6 numbers, than in short video clips with moving objects. In contrast, operating on raw pixels is expected to introduce more difficulty for machine learning models in terms of disentangling representations[85]. Future work could examine whether neural nets segment visual stimuli in a similar way as humans in grounded language learning.

In conclusion, our findings shed light on the relationship between language and language-learning systems, showing that linguistic structure is crucial for language learnability and systematic generalization in neural networks. Our results suggest that more structured languages are easier to learn, regardless of the learning system: a human, a recurrent neural network, or a large language model. Thus, generalization capabilities are heavily influenced by compositional structure, with both biological and artificial learning systems benefitting from more structured input by facilitating more systematic and transparent generalizations. In future work, we will analyze how this learnability bias for more structure affects neural networks engaged in collaborative communication games, and test how this kind of systematic structure arises in the first place in emergent communication simulations. Moreover, our findings give a clear prediction that children would benefit from more structure in the linguistic input, which we will test by conducting a learnability study with children.

## Methods

### Input languages
The input languages with different degrees of compositional structure come from a previous communication study in which groups of interacting participants took turns producing and guessing labels for different dynamic scenes, creating new artificial languages over time[68]. Ten of the final languages created by these groups then served as input languages for a follow-up study on language learnability with humans[28]. For our experiments, we used the same ten input languages. These input languages are considered the ground truth. Each of the ten input languages contains a set of 23 label-scene mappings. Each scene comprises one of four different shapes moving in different directions between 0 and 360 degrees. The languages vary in their degree of compositional structure, with structure scores ranging from 0.09 to 0.85.

### Topographic similarity to quantify compositional structure
Crucially, the ten input languages have different degrees of structure, ranging from languages with no structure to languages with consistent, systematic grammar. Each language has a structure score represented by topographic similarity[15], quantifying the degree to which similar labels describe similar meanings. The topographic similarity is measured as the Pearson correlation between all labels' pairwise length-normalized edit distances and their corresponding pairwise semantic differences. The semantic difference between two scenes is calculated as the sum of the difference in shape and the difference in angles[28]. The difference in shape is zero if the two scenes contain the same shape, and one otherwise. The difference in angles is

calculated as the absolute difference divided by 180. The topographic similarity of a language is then calculated as the pairwise correlation between all semantic differences and all normalized edit distances. For a complete list of input languages and their structure scores, see Supplementary Table 1.

### Human learning data
Aside from the input languages, we use reference data from 100 human participants learning these input languages[28]. The participants were different from those who created the languages. A hundred participants, ten per input language, engaged in repeated learning blocks consisting of passive exposure (in which the target label-meaning mappings were presented on the screen one by one), guessing trials (in which participants needed to pick the right scene from a set of possible distractors), and production trials (in which participants needed to generate a descriptive label for a target scene based on what they had learned). During training, humans received feedback on their performance.

### Large language models
For the large language models, we supplied the full training data of the respective input language to GPT-3.5: 23 lines consisting of shape-angle pairs in a textual format, and the corresponding target label. These 23 lines were followed by a single line that only contained shape and angle but no word. GPT-3.5 was made to predict the most likely word as completion, for which it could take into account the 23 triples presented in the prompt. In the memorization task, the target word appears earlier in the prompt, which means that the perfect solution would be to simply copy this word. In the generalization task, we gave GPT-3.5 a combination of shape and angle not present in the training data (and not in the prompt). The model generated the most likely descriptive word for the new shape-angle pair.

We had to make certain technical choices when using GPT-3.5. First, we chose a consistent input representation (Javascript Object Notation). We do not insert a task description to avoid potential bias. Instead we purely rely on next-token prediction. Second, we set the sampling temperature to zero, which controls the randomness of the generation, such that we obtain deterministic generations. Third, we do not impose any restrictions on the characters that can be generated but rely on its ability to detect this pattern from the training data. Fourth, we do not feed back GPT-3.5's previous productions into the prompt. Lastly, GPT-3.5's tokenization procedure (how text is split into subword tokens) could have been problematic for applying it to our artificial languages. However, we found that GPT-3.5 still reaches high memorization performance, which suggests that tokenization is not a problem. We have confirmed that the words of the artificial languages are tokenized as expected with OpenAI's Tokenizer (https://platform.openai.com/tokenizer): falling back to one token per character.

### Custom recurrent neural network architecture
Our custom model architecture (see Fig. 1, right) is based on two components: a generative component and a contrastive component. The generative component is conditioned on the input scene and generates a label letter by letter. The contrastive component ensures that the matching scenes and labels are close in the representation space and non-matching pairs are apart from each other. For processing the sequence of letters, each component uses a recurrent neural network, for which we use the well-known long short-term memory (LSTM)[72]. In the following, we describe the input representation before we describe the two components and their interactions. The models were implemented in PyTorch[91], version 2.3.

Scenes were shown to human participants as short videos[28]. For the recurrent neural networks, we use a simplified representation of the scenes. The rationale for choosing this input representation over images is that both humans and models receive the respective easiest

possible input type to process, allowing for a fair comparison[92,93]. For the recurrent neural networks, we employ a one-hot encoding of the shape concatenated with a sine and a cosine transformation of the angle. The sine−cosine transformation promotes a similar treatment of angles that are close to each other, while each unique angle can be distinguished. For example, shape 2 (between 1 and 4) moving at a 90-degree angle is converted to a vector (0, 1, 0, 0, 1, 0), shape 3 with 45 degrees is converted to (0, 0, 1, 0, 0.71, 0.71), and shape 4 with 135 degrees is converted to (0, 0, 0, 1, 0.71, − 0.71). We refer to the resulting 6-dimensional vector representation of the input as a scene **x**. By using this input representation, we focus on the ability of systematic generalization in language learning rather than the ability to learn disentangled representations. If the neural networks were trained on pixel input instead, the task would be more challenging as neural networks would need to learn disentangled representations on the fly[85].

Within the generative component, the input scene **x** is first encoded to a latent representation **h** by ENCODER, a feedforward network (we use a multilayer perceptron with one hidden layer), such that we obtain a latent representation **h** = ENCODER(**x**). This latent representation **h** is then used as the initial state of the recurrent neural network WRITER. The WRITER sequentially produces a sequence of letters, i. e., a label, as output. This WRITER consists of three modules: an input embedding for previously produced characters, an LSTM cell, and an output layer that produces the next letter.

For the contrastive component, we use another recurrent module READER that reads a label **m** sequentially (i. e., letter by letter) while updating its state. As for the WRITER, we again use an LSTM. A fully-connected layer transforms the final state into a latent representation **z**, such that **z** = READER(**m**), where **m** is the input label. The reading component is used for contrastive learning, i. e., they are trained so that the hidden representation of the label **z** matches the representation of the corresponding scene **h** = ENCODER(**x**), which is used as the initial hidden state of the generative WRITER module.

To ensure that the contrastive training procedure affects the generative component, we couple the two components: First, the embedding (i. e., the mapping between the agent's alphabet and the first latent representation) parameters are shared between the input layer of READER, the input layer of WRITER, and the output layer of WRITER. Second, the same encoder module is used in both the generative and the contrastive components (see Fig. 1).

The output dimension of ENCODER, the hidden state sizes of READER and WRITER, and the embedding size are all set to 50. A sensitivity analysis of the hidden size on the dependent variables of interest is provided in Supplementary Figs. 13–15. Similarly to Nakkiran et al.[59], larger hidden sizes led to a faster increase in memorization and generalization.

## Training procedure

We train the recurrent neural networks for multiple training rounds as in the experiments with human participants[28]. Each training round consists of three blocks: exposure, guessing, and production block, described in detail in the following. As typical in neural network training, we train the network with backpropagation and stochastic gradient descent, where the gradient is estimated based on a small number of examples (minibatches)[94,95]. The batch size, which also determines the number of distractors, is set to 5, reflecting human short-term memory constraints[96]. Only in the guessing block, we set the batch size to 1 and use the same distractors as in the experiments with human participants, instead of other exemplars from the same batch.

In the exposure block, human participants were exposed to scenes with the corresponding target labels. Therefore, we train the deep learning models using a loss function with two terms: a generative and a contrastive loss term. The generative loss, $\mathcal{L}_{gen}$, is a token-wise cross-entropy with the ground-truth label of the original language. The contrastive loss, $\mathcal{L}_{con}$, promotes similar latent representations of scenes and labels that correspond to each other and contrasts representations that do not. Specifically, we use the normalized temperature-scaled cross-entropy loss (NTXent)[73]. We use other scenes in the same batch as distractors for the contrastive loss term. The final loss function is $\mathcal{L} = \mathcal{L}_{gen} + \alpha_{con}\mathcal{L}_{con}$. The factor $\alpha_{con}$ determines the relative weight of the loss terms. For the main experiment, we use $\alpha_{con} = 0.1$. A sensitivity analysis using other values for $\alpha_{con}$ is provided in Supplementary Figs. 16–18.

In the guessing block, we use the same loss function as in the exposure block. The contrastive loss term $\mathcal{L}_{con}$ mirrors the task in which human participants had to select the correct scene against the distractors given a label. The generative loss term $\mathcal{L}_{gen}$ is used so that the model does not "forget" how to generate[97]. Notably, the guessing task itself could be also carried out by having the models generate a descriptive label for each scene and then select the closest one to the given label in terms of edit distance. However, we opted for optimizing shared parameters through a contrastive loss to ensure that the guessing task would also have an effect on the production task (and vice-versa).

In more detail, the latent representation **z** = ENCODER(**x**) of the scene **x** should be closest to the latent representation **z′** = READER(**m**) of the corresponding label **m**. The difference from exposure training is that in the guessing block, we use the identical distractors used in experiments with humans, whereas, in the exposure block, we use the other scenes from the same batch. The trajectory of guessing accuracy during training is shown in Supplementary Fig. 7.

In the production block, a scene was presented to human participants, who had to produce a label. We again use the same generative loss as in the previous block, $\mathcal{L}_{gen}$, to model the production block. In the production block, however, we omit the contrastive loss term and train only on generation. Thus, the loss function for the production block is $\mathcal{L} = \mathcal{L}_{gen}$.

The parameters are randomly initialized by He initialization[98], the default initialization method in PyTorch. We employ the widely used Adam optimizer[99] to carry out the optimization of the loss function with the default learning rate of $10^{-3}$. As common in machine learning, we have to make certain decisions about the neural network architecture design, optimization procedure, and hyperparameters. All these decisions may impact the results. However, we have varied relevant hyperparameter settings and found that the results are robust and do not dependent on specific settings of the hyperparameters (see Supplementary Methods and Supplementary Figs. 19 and 20).

## Measures

Production similarity measures the overlap between two sets of labels. It is computed as one minus the normalized edit distance between pairs of labels. For our analysis, we use production similarity once to quantify the similarity between the generated labels and the ground truth of the input languages, and once to quantify the similarity of labels generated by neural network agents with labels produced by human learners. For example, a recurrent neural network that produced 'muif-a' for shape 3 moving in direction 360 degrees would have a high production similarity to the majority of human participants who produced 'muif-i'.

The generalization score measures the degree of systematicity during the generalization test[28]. We take two sets of scenes: a training set, on which the agents were trained, and a test set, on which the agents were not trained. We then do the following for each agent. First, we take two sets of labels: one previously generated for each training scene by the agent and another that we let the agent generate for each test scene. Second, the difference between train and test scenes is measured by pairwise semantic difference. Semantic difference is calculated as in topographic similarity. Third, the difference between generated labels for the train and test scenes is measured by pairwise normalized edit

distance. Finally, we compute the Pearson correlation between these two differences across all scenes. Then, we take the average correlation coefficient across all agents as the generalization score.

The convergence score measures the similarity in the generalization test between agents that learned the same language. We take the test set on which the agents have not been trained and let each agent produce a label for each scene. We compute the pairwise normalized edit distance between all generated labels per scene so that if we have $n$ test scenes and $k$ agents, we compute $n \cdot \frac{k(k-1)}{2}$ distances. We then compute the average distance across both scenes and labels and take one minus the average distance as the convergence score. Therefore, if all agents produce the same label for each test scene, we would get a convergence score of 1. Conversely, if each agent produced a different label for the same scene, the convergence score would be zero.

## Statistical analyses

We trained 100 differently-initialized neural network models over 100 rounds for each of the ten input languages. The testing in each round consisted of 23 memorization and 13 generalization examples. This makes a total of 2.3M memorization and 1.3M generalization test results subject to statistical analyses. Significance was tested using linear mixed-effects models, as implemented in the Python package statsmodels[100] (version 0.13), for production similarity (LME 1), generalization score (LME 3), generalization convergence (LME 4), as well as production similarity to humans in memorization (LME 2) and generalization (LME 5). We use the structure score and the logarithmized round number in all measures as a fixed effect. The number of rounds was logarithmized following scaling laws of neural language models[60]. Both the structure score and the logarithmized round number were centered and scaled. We consider two random effects: the random seed for initialization (which also determines the input language) and the specific scene. Normality was tested via QQ-plots on the residuals. For LME 5, scaling the log-transformed round number to unit variance hindered convergence, so the log rounds were only centered. The full results of the statistical models are provided in Supplementary Table 2, with partial regression plots shown in Supplementary Figs. 8–12. In Supplementary Table 3, we provide an additional analysis of production similarity to ground truth at rounds 10, 40, 70, and 100.

## Reporting summary

Further information on research design is available in the Nature Portfolio Reporting Summary linked to this article.

# Data availability

The input data used in this study are available in the OSF database under accession code https://osf.io/d5ty7/[101]. The results data generated in this study have been deposited in the Zenodo database under accession code https://doi.org/10.5281/zenodo.14205452[102].

# Code availability

The source code for reproducing our experiments has been deposited in the GitHub database under accession code https://github.com/lgalke/easy2deeplearn[103].

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

## Acknowledgements
We thank Dota Tianai Dong, Koen de Reus, Yosef Prat, Tal Simon, Willem Zuidema, Tessa Verhoef, Mitja Nikolaus, Marieke Woensdregt, and Adam Kohan for their valuable comments and discussions. We thank Shinje Wu and Marianne de Heer Kloots for sharing their data.

## Author contributions
Contributions listed according to CRediT – Contributor Roles Taxonomy (https://credit.niso.org): L.G. contributed conceptualization, data curation, formal analysis, investigation, methodology, software, validation, visualization, writing – original draft, and writing – review and editing. Y.R. contributed conceptualization, methodology, supervision, and writing – review and editing. L.R. contributed conceptualization, methodology, resources, supervision, writing – original draft, and writing – review and editing.

## Funding

## Competing interests
The authors declare no competing interests.
