## [Transparent Peer Review file · Nature Communications]

Deep neural networks and humans both benefit from compositional language structure

Corresponding Author: Professor Lukas Galke

Version 0:

Reviewer comments:

Reviewer #1

(Remarks to the Author)

This manuscript presents a detailed comparison of learning and generalization behavior in artificial languages between adult humans and two different deep neural networks (DNNs): the instruction-tuned Transformer-based LLM GPT-3.5, and a custom RNN-based architecture designed to emulate the human experimental training procedure. For readers from an NLP or ML background, the 10 artificial languages used in this study will be surprising. While compositionality in the NLP literature is typically assessed via procedurally generated formal languages which are highly structured by definition (e.g. SCAN, COGs, etc.), the artificial languages in this experiment arise from an earlier round of iterated learning experiments with human participants. These languages are an interesting hybrid --- they're artificially constrained in vocabulary and scope due to their experimental origins, but nonetheless display some naturalistic variation in their degree of compositionality, as expressed through the somewhat opaque Structure Score. A previous study with these languages showed that adult human speakers are highly sensitive to the Structure Score: they learn faster, make fewer memorization errors, and converge on similar generalization patterns if trained on a language with more compositional structure. The key research question addressed in this paper is whether DNNs show similarly structure-sensitive benefits in language learning. The main finding --- conveyed in a very thorough analysis of Structure Score's impact on a range of measures --- is yes, they do.

This paper is very strong overall. The authors situate their work at the intersection of two broader literatures in cognitive science and NLP, but scrupulously restrict the scope of their investigation and claims, with the result that their findings are well-substantiated by the evidence. The detailed, targeted comparison to human performance on the same task is particularly noteworthy against the backdrop of an NLP literature where the term "human-like" is often extended without appropriate experimental justification. Moreover, the findings are interesting and noteworthy --- both GPT-3.5 and the RNNs show consistently human-like sensitivity to compositional structure in the linguistic input. This effect appears robust not only in direction (which the authors confirm with statistical tests), but also often in magnitude. While this finding is quite interesting in the case of GPT-3.5, it becomes especially convincing in light of the custom RNNs which are trained only on the same experimental data provided as input to human participants. In general, we're convinced by the rigor of experimentation and analysis shown here.

Our main concern is in the paper framing, which we think is both over- and under-adapted to a broader ML/NLP audience. Over-adapted in the sense that the introduction motivates this work with respect to two strands within the NLP literature --- compositional generalization and language emergence --- but does not address the core concerns of either. As far as we can tell, the compositional generalization studied in this experiment involves previous unseen combinations of features where each feature has been seen in a range of other combinations in the training data (though if this understanding is incorrect, we welcome the authors' clarification). While in-domain generalization of this type is certainly important, it doesn't necessarily bear on the out-of-domain generalization central to this literature. Consider for example COGs (Kim and Linzen, EMNLP 2020), which assesses lexical generalization (observing a feature in only one context, and generalizing it to a broader range of contexts) and structural generalization (parsing a sequence longer than any observed in training); systematic generalization to unseen feature combinations as studied here is a necessary but not sufficient precondition of such behavior. Similarly, we don't think that the ability of neural nets to learn structured languages gives us much insight into why they fail to produce compositionally structured codes in language emergence experiments. The title is instructive here: very few people in NLP likely need convincing that "DNNs are Structured Language Learners", given that augmentation schemes to produce structured input data are currently the most robust approach to inducing NN compositional

generalization in practice (e.g. Akyürek and Andreas, ACL 2023). This is not to say that NLP researchers won't find the current paper interesting and valuable, but we think a more representative framing of the contents would make its value clearer. For example, a title like "DNNs and Humans Generalize Exactly the Structure in their Input and No More" or "DNNs and Humans Are Similarly Responsive to Compositional Structure in Language Learning" would highlight what we perceive as the paper's main contribution.

We also find this paper somewhat *under-*adapted to a broader audience, in the sense that a reader who lacks familiarity with artificial language learning studies will struggle mightily to follow the research presented here. The value of these results would be more obvious if preceded and accompanied by illustrative examples from the test languages. Some things that a reader might not expect at the outset, and would benefit from having clearly explained: a) there are ten languages; b) they emerge from an experimental setting, so the range of compositional structure is in some sense "naturalistic" relative to the formal languages common in the NLP literature; c) the ten languages emerged in separate contexts, and so do not necessarily have anything to do with each other - this isn't one language which has been manipulated to systematically vary its Structure Score.

Our second concern regards the similarity to humans. One of the key results (Figure 3) indicates that the models generalize in a way more similar to human participants when the input language is more structured. A priori, two things could possibly be going on: One possibility is that both models and humans generalize in arbitrary unpredictable ways when the input language is unstructured; in this case, the similarity between different humans will be similarly low as the similarity between models and humans. Another possibility is that the humans, when exposed to the unstructured languages, generalize in predictable ways that are different from what the models do. In this case, the similarity between different humans will be higher than the similarity between models and humans. The paper appears to suggest the first interpretation, but it would be easy to explicitly distinguish between the two by reporting the similarities between the different humans for the different languages.

Some concrete suggestions regarding clarity and framing:

- Give more examples of the artificial languages to give readers an intuitive sense of their structure and predictability. Table 1 is an excellent start, but could be expanded --- for instance, including relevant instances of the training data, which is shared across humans and the language models. Also, the current design of Table 1 doesn't indicate to what extent these examples are representative. Would it be feasible to include more examples in the SI?
- Examples could also help explain unfamiliar measures, e.g. "an RNN who produced 'muif-a' for 3 360 would have high production similarity to the majority of participants who produced 'muif-i'"
- Another place where examples might help: figure captions, as figures sometimes end up far from the place where they're discussed in text. Examples could also clarify the referents for the data points represented in each figure. e.g. in Fig. 2, we infer the ten points in (b) to correspond to the ten input languages, and in (a) each point is a human tested on one of the ten languages - but in Fig. 3a, each point is a generalization item? and in Fig. 4b, each point is a memorization error? Examples would be really helpful here.
- Clarify different measures --- for example, call the y-axis "Human Label Similarity" in Fig. 3, and "True Label Similarity" in Fig. 4.
- As discussed above, it would be instructive to see how similar human generalizations are to each other in Figure 3, as a baseline/context for the depicted DNN similarity to humans.
- A final request: GPT-3 is a pure language model, and GPT-3.5 has extensive additional instruction tuning and RLHF. This is a very significant difference, and we implore the authors to make it clear by consistently referring to GPT-3.5 rather than GPT-3 in figures and paper text.

With all that said, we think the findings here are impactful for both NLP and cognitive science, and would encourage the authors to clarify the framing and experimental design so that readers can appreciate the results. The authors note that the similarity between human and DNN in terms of structure-sensitivity motivates DNNs as cognitive models --- at least for artificial language learning experiments such as this one. This is a well-motivated point, but we are also intrigued by the apparent discrepancies, e.g. in Figure 4, where human memory performance spans a much broader range. In light of the language evolution theme, it also seems noteworthy that neither humans nor models appear to impose additional compositional structure on the learned languages. In Fig. 2, DNNs appear fairly consistently at the upper range for systematic generalization, which suggests that they successfully detect and exploit compositional patterns, perhaps even more consistently and successfully than humans on the whole; however, they do not appear to regularize or increase predictability within the language --- both humans and DNNs use the structure in the input, and no more. It's good to know that they're similarly sensitive to linguistic structure, but we'll need more research to determine how that structure arises in the first place.

Reviewer #3

(Remarks to the Author)

Reviewer #4

(Remarks to the Author)

This article compares how humans and models learn artificial languages that vary in degrees of compositional structure. For both people and neural networks, learning a more structured language results in more structured generalizations. Additionally, people and neural nets produced more similar labels when generalizing in more structured languages. This was found both in a GPT-3 model performing in-context learning and a custom recurrent neural network.

I like this article and would be glad to see a version published in Nature Communications. The authors chose a compelling paradigm in comparing humans and neural nets side-by-side on the same language learning tasks. The results are analyzed in depth. I appreciate that the authors tested both GPT-3 and a custom neural network model. The findings are significant in that they address longstanding questions about differences in how people and neural networks can learn and generalize structure in language.

My main worry is that I can't discern whether these findings tell us something about the nature of neural networks versus any other capable learner. Indeed, after learning an unstructured language, there would be little basis for generalization when considering a novel input, regardless of what kind of learner one is. With this in mind, I am not sure how impressed I should be with the comparisons in Figures 2 and 3, showing how more structured input leads to more structured generalizations. Figures 4 and 5 examine errors in learning the training examples, which indeed is different and perhaps a signal of the networks exploiting compositional structure during the learning process. However without additional baselines and/or discussion of this issue, it's difficult to interpret these results. Is it possible for a model to show the opposite pattern: less structured inputs lead to more structured generalizations?

Other comments:

1) There are a number of custom scoring metrics for the models that are difficult to understand and keep track of: "generalization score", "structure score", "production score", etc. I kept flipping back and forth between the main text, Methods section, and ref. 27 to try to understand and keep track of them. I suggest a glossary in the Methods with worked examples for additional clarity.

2) I had trouble understanding the motivation for needing both the generative and contrastive component of the custom recurrent neural net (RNN). The article says... "while a contrastive component shapes the latent space and enables the models to carry out guessing tasks during learning (i.e., given a label, pick the correct scene from a set of distractors)" (pg. 6). However, it seems the generative component could also perform this function as follows for the given 'label' and a set of different scenes 'S':
$$\operatorname{argmax}_{s \in S} p(\text{label} | s),$$
such that $p(\text{label} | s)$ is done by evaluating the generative component. Was this alternative tried? More information and motivation is needed for the architectures.

3) For Figure 4, when is the training is the error rate calculated? As the authors state in the paper, RNN should eventually be able to learn each of these languages, but in this plot they are showing errors. Also, why is GPT-3 making mistakes here if it is just a copying task?

4) For Fig. 5, why do languages with a middle amount of structure (mid-low, mid, mid-high) often have an advantage in early learning? This should be discussed.

5) In Fig. 4, the models don't seem to fit the human data very closely. This should be discussed too.

6) I would be nice, although maybe not essential, to see a proof of concept that the custom RNN model can work from raw images too. The highly-processed symbolic input seems like a step back from where neural network modeling is today. Thus, I suggest a supplemental simulation where the model processes the raw image stimuli, e.g., through a pre-trained image encoder.

7) "Semantic difference is calculated as described above under Input Languages" (pg. 16). I couldn't find a description of this calculation, and instead went back to ref. 27.

Version 1:

Reviewer comments:

Reviewer #1

(Remarks to the Author)

Thank you to the authors for this detailed rebuttal. The proposed changes address all of our substantial concerns, and make a more clear and compelling case for the authors' interpretation of these modeling results.

We have only one more minor recommendation, which is that the discussion on compositionality and variation would be strengthened by considering or at least referring to this relevant recent ICLR citation: <https://openreview.net/pdf?id=Yzz6vIX7V>- (Conklin and Smith, Compositionality with Variation Reliably Emerges in Neural Networks, ICLR 2023). We would also recommend a final review for consistent formatting (e.g. opening and closing quotation marks, single and double quotes) and typos.

Aside from these minor points, we look forward to the final publication of this submission.

Reviewer #3

(Remarks to the Author)

Reviewer #4

(Remarks to the Author)

I am satisfied with the revisions and responses to my comments. I would be happy to see the article accepted.

NCOMMS-24-13508-T Author Response to Reviews

Dear Editor, Dear Reviewers,

We appreciate the positive comments and the suggestions for improving the manuscript are extremely valuable. The revised manuscript indicates **added text in blue**. We describe how we improve the paper based on the comments in our point-by-point response below.

Best wishes,
The authors

Reviewer 1+3

This manuscript presents a detailed comparison of learning and generalization behavior in artificial languages between adult humans and two different deep neural networks (DNNs): the instruction-tuned Transformer-based LLM GPT-3.5, and a custom RNN-based architecture designed to emulate the human experimental training procedure. For readers from an NLP or ML background, the 10 artificial languages used in this study will be surprising. While compositionality in the NLP literature is typically assessed via procedurally generated formal languages which are highly structured by definition (e.g. SCAN, COGs, etc.), the artificial languages in this experiment arise from an earlier round of iterated learning experiments with human participants. These languages are an interesting hybrid --- they're artificially constrained in vocabulary and scope due to their experimental origins, but nonetheless display some naturalistic variation in their degree of compositionality, as expressed through the somewhat opaque Structure Score. A previous study with these languages showed that adult human speakers are highly sensitive to the Structure Score: they learn faster, make fewer memorization errors, and converge on similar generalization patterns if trained on a language with more compositional structure. The key research question addressed in this paper is whether DNNs show similarly structure-sensitive benefits in language learning. The main finding --- conveyed in a very thorough analysis of Structure Score's impact on a range of measures --- is yes, they do.

This paper is very strong overall. The authors situate their work at the intersection of two broader literatures in cognitive science and NLP, but scrupulously restrict the scope of their investigation and claims, with the result that their findings are well-substantiated by the evidence. The detailed, targeted comparison to human performance on the same task is particularly noteworthy against the backdrop of an NLP literature where the term "human-like" is often extended without appropriate experimental justification. Moreover, the findings are interesting and noteworthy --- both GPT-3.5 and the RNNs show consistently human-like

sensitivity to compositional structure in the linguistic input. This effect appears robust not only in direction (which the authors confirm with statistical tests), but also often in magnitude. While this finding is quite interesting in the case of GPT-3.5, it becomes especially convincing in light of the custom RNNs which are trained only on the same experimental data provided as input to human participants. In general, we're convinced by the rigor of experimentation and analysis shown here.

Thank you for these very positive remarks!

Our main concern is in the paper framing, which we think is both over- and under-adapted to a broader ML/NLP audience. Over- adapted in the sense that the introduction motivates this work with respect to two strands within the NLP literature --- compositional generalization and language emergence --- but does not address the core concerns of either. As far as we can tell, the compositional generalization studied in this experiment involves previous unseen combinations of features where each feature has been seen in a range of other combinations in the training data (though if this understanding is incorrect, we welcome the authors' clarification). While in-domain generalization of this type is certainly important, it doesn't necessarily bear on the out-of-domain generalization central to this literature. Consider for example COGs (Kim and Linzen, EMNLP 2020), which assesses lexical generalization (observing a feature in only one context, and generalizing it to a broader range of contexts) and structural generalization (parsing a sequence longer than any observed in training); systematic generalization to unseen feature combinations as studied here is a necessary but not sufficient precondition of such behavior.

Thank you for pointing out this important distinction. We have now clarified the distinction between in-domain generalization and out-of-domain generalization in the Discussion of the revised manuscript (L308-311):

“Our findings have further implications for machine learning, where systematic generalization beyond the training distribution (out-of-domain) is of high interest [refs – including COGs and SCAN]. Systematic in-domain generalization, as studied here, is a critical prerequisite for systematic out-of-domain generalization.”

Similarly, we don't think that the ability of neural nets to learn structured languages gives us much insight into why they fail to produce compositionally structured codes in language emergence experiments. The title is instructive here: very few people in NLP likely need convincing that "DNNs are Structured Language Learners", given that augmentation schemes to produce structured input data are currently the most robust approach to inducing NN compositional generalization in practice (e.g. Akyürek and Andreas, ACL 2023). This is not to say that NLP researchers won't find the current paper interesting and valuable, but we think a more representative framing of the contents would make its value clearer. For example, a title like "DNNs and Humans Generalize Exactly the Structure in their Input and No More" or "DNNs and Humans Are Similarly Responsive to Compositional Structure in Language Learning" would highlight what we perceive as the paper's main contribution.

Thank you for sharing your reading and recommendations. Based on your suggestion for the title, we have changed it to: “What makes a language easy to deep-learn? Deep neural networks and humans similarly benefit from compositional structure”. We have further added the suggested reference in the Intro.

We also find this paper somewhat *under-*adapted to a broader audience, in the sense that a reader who lacks familiarity with artificial language learning studies will struggle mightily to follow the research presented here. The value of these results would be more obvious if preceded and accompanied by illustrative examples from the test languages. Some things that a reader might not expect at the outset, and would benefit from having clearly explained: a) there are ten languages; b) they emerge from an experimental setting, so the range of compositional structure is in some sense "naturalistic" relative to the formal languages common in the NLP literature; c) the ten languages emerged in separate contexts, and so do not necessarily have anything to do with each other - this isn't one language which has been manipulated to systematically vary its Structure Score.

Thank you for this suggestion, we have added the following to the last paragraph of the Introduction (L98-106) and we have revised Figure 1 to better illustrate the input languages and their difference in structure (also copied here):

“We consider 10 input languages, each of which has emerged independently and spontaneously through a group communication experiment with adult human participants (Raviv et al., 2019). The languages describe four different novel shapes moving on the screen in a different direction (0-360 degree), and vary in their degree of compositional structure: ranging from fully idiosyncratic languages with entirely different labels for two related meaning (e.g., 'kuim' and 'goom' for the same shape moving into a different direction) to highly structured languages, which re-use parts of the descriptive label (e.g., referring to the two scenes as 'fest-ii' and 'fest-ui'). See Figure 1.”

Figure 1: Overview of input languages (*Top*), the experimental procedure (*Bottom Center*) along with exemplary input data from one language (*Bottom Left*), and the model architecture (*Bottom Right*). Low-structured input languages show no signs of systematicity or compositionality, whereas high-structured languages are systematic and compositional with respect to both attributes: shape and angle. For each language, we train the model for multiple rounds of exposure, guessing, production. After each round, we conduct a memorization test to evaluate productions for previously seen items and a generalization test evaluating the productions for new items. Elements in the upper part are re-used and adapted with permission from Raviv et al. [67].

Our second concern regards the similarity to humans. One of the key results (Figure 3) indicates that the models generalize in a way more similar to human participants when the input language is more structured. A priori, two things could possibly be going on: One possibility is that both models and humans generalize in arbitrary unpredictable ways when the input language is unstructured; in this case, the similarity between different humans will be similarly low as the similarity between models and humans. Another possibility is that the humans, when exposed to the unstructured languages, generalize in predictable ways that are different from what the models do. In this case, the similarity between different humans will be higher than the similarity

between models and humans. The paper appears to suggest the first interpretation, but it would be easy to explicitly distinguish between the two by reporting the similarities between the different humans for the different languages.

Thank you for pointing out this nuance. We believe the reason for this trend is that the ways to generalize are more clearly defined in highly structured languages, while there are no (or less transparent) generalization patterns in low and medium structured language -, leading both humans and neural nets to a higher variation. In other words, there are more equally-reasonable options in low-structured languages. This is in agreement with the first possibility you mentioned and also supported by the results obtained in previous work from humans, showing increased convergence between participants when learning higher structured languages (Raviv et al. 2021). To further clarify this point, we have added the requested plot of human-to-human similarity to Figure 3 as subplot (A) (see below). We have also re-evaluated the RNNs similarity to humans for consistency among the 3 subplots (now: *average to all* human productions on the same language/item).. We have further added a paragraph at the beginning of the Discussion to explain this interpretation of the results (L275-290):

“Specifically, we find very similar effects of structure on generalization and on the similarity to humans across all three learning systems. While we find a different slope for humans and RNNs in the memorization error analysis (likely due to RNNs being less impacted by memorization difficulty given sufficient training), the general trend is consistent: for both humans and artificial agents, exposure to more structured languages leads to production errors that are nevertheless more similar to the correct labels (i.e., their errors are less “wrong”).”

“We assume that the reason for the increased similarity between machines and humans is that the ways to generalize are more transparent in high-structured languages, while there are none or less transparent generalization patterns available in low- and medium-structured languages. This leads both humans and neural networks to a higher production variation in lower structured languages, as different options on how to generalize are equally likely. This point is well supported by results from humans, who indeed show increased convergence between participants when learning higher structured languages [Raviv et al. 2021]. Our results thereby demonstrate that what is more transparent for humans is also more transparent for deep neural networks.”

The new plot is also shown below:

Figure 3: **Final Similarity to Humans during Generalization** Final production similarity with (other) human participants during generalization achieved by humans (A), GPT-3.5 (B) and recurrent neural networks (C) for each of the input languages. The x-axis shows the structure score of the input languages. Each point displays the production similarity (length-normalized edit distance) between human productions and the model’s generated label for an item. For example, a recurrent neural network that produced ‘muif-a’ for shape 3 moving in direction 360 degree would have a high production similarity to the majority of human participants who produced ‘muif-i’. Error regions show 95% confidence intervals estimated via bootstrapping. More structure in the input language leads to more similarity to human participants for both RNNs and GPT-3.5.

Some concrete suggestions regarding clarity and framing:

- Give more examples of the artificial languages to give readers an intuitive sense of their structure and predictability. Table 1 is an excellent start, but could be expanded --- for instance, including relevant instances of the training data, which is shared across humans and the language models. Also, the current design of Table 1 doesn't indicate to what extent these examples are representative. Would it be feasible to include more examples in the SI?

Examples in Table 1 were selected to be representative of the differently-structured input languages. We have further revised Fig. 1 (as shown above) to better illustrate the different levels of structure. Additionally, we have added more examples to the SI, stratified over similarity to humans, and added a pointer to the paper "(more examples are provided in Tab. 4 and 5 of the SI)". Furthermore, we have added the full raw data as a Supplementary Files.

- Examples could also help explain unfamiliar measures, e.g. "an RNN who produced 'muif-a' for 3 360 would have high production similarity to the majority of participants who produced 'muif-i'"

Thank you for this suggestion. We have added this example to the Measures section (L534-536):

For example, a recurrent neural network that produced 'muif-a' for shape 3 moving in direction 360 degrees would have a high production similarity to the majority of human participants who produced 'muif-i'.

In addition, we added a tabular Glossary of the Metrics used in the current study for added clarity (see response to Reviewer 4 below) – and added one example to the captions of Figs. 3, 4, and 5 (next point).

- Another place where examples might help: figure captions, as figures sometimes end up far from the place where they're discussed in text. Examples could also clarify the referents for the data points represented in each figure. e.g. in Fig. 2, we infer the ten points in (b) to correspond to the ten input languages, and in (a) each point is a human tested on one of the ten languages - but in Fig. 3a, each point is a generalization item? and in Fig. 4b, each point is a memorization error? Examples would be really helpful here.

Thank you for pointing this out. Indeed, in Fig. 3 and 4., each point represents a single item in the language, whereas in Fig. 2 each point represents the generalization score calculated for the entire input language (which cannot be calculated per-item). We have updated the captions of Figures 2, 3, and 4 to make this point more transparent – and also added examples to the figure captions as suggested:

Fig 2: "[...] Each point corresponds to the generalization score calculated for the entire input language. This score reflects the degree to which learners systematically generalized new labels relative to the labels they learned. For example, generalization score would be high if learners

successfully recombines previously used parts, e.g., combining 'muif' for the shape and 'i' for the direction into 'muif-i [...]'”

Fig 3: “[...] Each point corresponds to the production similarity score (calculated as length-normalized edit distance) between humans’ productions and models’ productions for every item in the language. For example, a recurrent neural network that produced 'muif-a' for shape 3 moving in direction 360 degrees would have a high production similarity to the majority of human participants who produced 'muif-i'. [...]

Fig 4: “[...] Each point corresponds to the production similarity score (calculated as length-normalized edit distance) between an erroneously memorized label for a given item and the correct corresponding label as it appears in the input language. For example, 'wangsus' has a higher similarity with 'wangsuus' than 'gempt' [...]"

- Clarify different measures --- for example, call the y-axis "Human Label Similarity" in Fig. 3, and "True Label Similarity" in Fig. 4.

We changed the axis labels for the measures and relabeled the y-axes as suggested. These item-level measures are now also shown in the new Glossary of Metrics table.

- As discussed above, it would be instructive to see how similar human generalizations are to each other in Figure 3, as a baseline/context for the depicted DNN similarity to humans.

We have added a third plot showing similarity between humans and other humans (see above).

- A final request: GPT-3 is a pure language model, and GPT-3.5 has extensive additional instruction tuning and RLHF. This is a very significant difference, and we implore the authors to make it clear by consistently referring to GPT-3.5 rather than GPT-3 in figures and paper text.

We agree that this distinction is important and have changed all occurrences of GPT-3 to GPT-3.5 in the text and figures.

With all that said, we think the findings here are impactful for both NLP and cognitive science, and would encourage the authors to clarify the framing and experimental design so that readers can appreciate the results. The authors note that the similarity between human and DNN in terms of structure-sensitivity motivates DNNs as cognitive models --- at least for artificial language learning experiments such as this one. This is a well-motivated point, but we are also intrigued by the apparent discrepancies, e.g. in Figure 4, where human memory performance spans a much broader range. In light of the language evolution theme, it also seems noteworthy that neither humans nor models appear to impose additional compositional structure on the learned languages. In Fig. 2, DNNs appear fairly consistently at the upper range for systematic generalization, which suggests that they successfully detect and exploit compositional patterns, perhaps even more consistently and successfully than humans on the whole; however, they do not appear to regularize or increase predictability within the language --- both humans and DNNs use the structure in the input, and no more. It's good to know that they're similarly

sensitive to linguistic structure, but we'll need more research to determine how that structure arises in the first place.

Thank you for the positive words and these suggestions. We emphasized this question for future work on how structure emerges in the first place (L348-349) and have added a new paragraph to the end of the Discussion, discussing potential training time effects (L343-352):

“In future work, we will analyze how this learnability bias for more structured languages affects neural networks engaged in collaborative communication games, and test how this kind of systematic structure arises in the first place in emergent communication simulations.”

“An interesting direction for future research is to investigate potential differences in the amount of training that a neural network needs compared to humans. Through anchoring our experiments in human data, we were able to directly identify the point during training at which recurrent neural networks equalize with human participants. However, the location of this point depends on various factors such as the amount of data, the number of parameters that are optimized, and the number of optimization steps – which makes it challenging to predict this point in advance. While we have identified this point through analyzing the learning trajectory, our analysis does not depend on it, as all measures including the similarity between humans and machines are calculated based on productions taken at the end of training.”

Code provides appropriate instructions. We were able to run the Python code.

We appreciate the reviewers' efforts to run the code and are glad that it could be run successfully.

Reviewer 4

This article compares how humans and models learn artificial languages that vary in degrees of compositional structure. For both people and neural networks, learning a more structured language results in more structured generalizations. Additionally, people and neural nets produced more similar labels when generalizing in more structured languages. This was found both in a GPT-3 model performing in-context learning and a custom recurrent neural network.

I like this article and would be glad to see a version published in Nature Communications. The authors chose a compelling paradigm in comparing humans and neural nets side-by-side on the same language learning tasks. The results are analyzed in depth. I appreciate that the authors tested both GPT-3 and a custom neural network model. The findings are significant in that they address longstanding questions about differences in how people and neural networks can learn and generalize structure in language.

Thank you for this positive and encouraging feedback.

My main worry is that I can't discern whether these findings tell us something about the nature of neural networks versus any other capable learner. Indeed, after learning an unstructured language, there would be little basis for generalization when considering a novel input,

regardless of what kind of learner one is. With this in mind, I am not sure how impressed I should be with the comparisons in Figures 2 and 3, showing how more structured input leads to more structured generalizations. Figures 4 and 5 examine errors in learning the training examples, which indeed is different and perhaps a signal of the networks exploiting compositional structure during the learning process. However without additional baselines and/or discussion of this issue, it's difficult to interpret these results. Is it possible for a model to show the opposite pattern: less structured inputs lead to more structured generalizations?

Thank you for sharing your thoughts. To some extent, we agree that the surprise factor may be limited, and that it indeed makes much sense that more structured input would lead to more structured generalization. However, given that previous research has suggested this may not be true for neural networks (e.g., Chaabouni et al. 2019), we think it was important to test this point under great scrutiny, and with direct comparison to human data.

Other comments:

1) There are a number of custom scoring metrics for the models that are difficult to understand and keep track of: "generalization score", "structure score", "production score", etc. I kept flipping back and forth between the main text, Methods section, and ref. 27 to try to understand and keep track of them. I suggest a glossary in the Methods with worked examples for additional clarity.

Thank you for this excellent suggestion. We have added a new Table “Glossary of Metrics” to the paper (copied here):

Table 1: Glossary of Metrics

Metric	Description
Production Similarity	One minus length-normalized edit distance
Semantic Difference	Sum of the difference in shape (1 if different and 0 otherwise) and the absolute difference in angles (divided by 180)
Structure Score	Correlation between semantic difference and length-normalized edit distance on all pairs in the input language
Generalization Score	Correlation between (a) pairwise semantic difference and (b) pairwise length-normalized edit distance, where (a) and (b) are calculated on all pairs between memorized labels and generalized labels
Convergence Score	Average production similarity for the same items between different learners trained on the same language
Human Label Similarity	Item-level production similarity to (other) human learners, averaged across different human learners
True Label Similarity	Item-level production similarity to input language

2) I had trouble understanding the motivation for needing both the generative and contrastive component of the custom recurrent neural net (RNN). The article says... "while a contrastive component shapes the latent space and enables the models to carry out guessing tasks during learning (i.e., given a label, pick the correct scene from a set of distractors)" (pg. 6). However, it seems the generative component could also perform this function as follows for the given 'label' and a set of different scenes 'S': $\text{argmax}_{\{s \in S\}} p(\text{label} | s)$, such that $p(\text{label} | s)$ is done by evaluating the generative component. Was this alternative tried? More information and motivation is needed for the architectures.

We agree that this suggested approach would in principle also facilitate guessing tasks. However, if we had done it this way, we would not have been able to backpropagate through the argmax - and as such the guessing task would have had no effect on production behavior. Since both the memorization and the generalization tests are production tasks, we opted for an approach in which the guessing task would have an influence on production through shared parameters. We have added an explanation for this choice in the methods section (L505-509):

"[...] Notably, the guessing task itself could be also carried out by having the models generate a descriptive label for each scene and then select the closest one to the given label in terms of edit distance. However, we opted for optimizing shared parameters through a contrastive loss to ensure that the guessing task would also have an effect on the production task (and vice-versa)."

3) For Figure 4, when is the training is the error rate calculated? As the authors state in the paper, RNN should eventually be able to learn each of these languages, but in this plot they are showing errors. Also, why is GPT-3 making mistakes here if it is just a copying task?

The error rate is the fraction of items on which the models/humans achieved less than 1.0 production similarity (length-normalized edit distance) with respect to the input language's "ground truth" label. The RNNs results were taken at the end of our training runs (precisely, at 100 epochs), where they almost reached perfect memorization performance on average, but still made some mistakes. Similarly, GPT-3.5's ability to copy the correct label, i.e. look up the correct shape/angle pair in its context window, is not perfect – at least in our setting without providing an explicit instruction to do so (as we did not want to bias the model in a certain direction). We have added a clarification of this to L261 ("Although all languages can eventually be (almost) perfectly learned") and to the caption of Fig 4 ("after 100 epochs of training").

4) For Fig. 5, why do languages with a middle amount of structure (mid-low, mid, mid-high) often have an advantage in early learning? This should be discussed.

Thank you for raising this question. We have added an explanation of this to the Discussion (L291-299):

"Analyzing the learning trajectory of recurrent neural networks, we find that languages with mid and mid-low structures often show an advantage in both memorization and generalization during the

early stages of learning. This may be due to the fact that these mid-structured languages trade off full expressiveness with more simplicity (see Tab. 1 of the SI) . For example, one of the mid-structured languages includes a marker for 'moving on the diagonal', but does not distinguish the direction of the movement (e.g., center to north-east vs. center to south-west). As a result, the same label is used for two distinct meanings, which is easier to learn in the first place (less variation), but not sufficient to fully differentiate between items and thus harming systematic generalization.”

The proportion of ambiguous labels can be found in the supplement.

5) In Fig. 4, the models don't seem to fit the human data very closely. This should be discussed too.

Thank you for pointing this out. We want to emphasize that the general trend is the same, but we agree that the slope differs more in this error analysis than in the other plots in the paper. It could be that this is because humans have more difficulty memorizing all 23 different labels when those are not systematic. It seems that this is less of an issue for the models though. However, the general trend that more structure leads to better memorization holds true for all three learning systems. We have added this point to the discussion (L275-281):

“Specifically, we find very similar effects of structure on generalization and on the similarity to humans across all three learning systems. While we find a different slope for humans and RNNs in the memorization error analysis (likely due to RNNs being less impacted by memorization difficulty given sufficient training), the general trend is consistent: for both humans and artificial agents, exposure to more structured languages leads to production errors that are nevertheless more similar to the correct labels (i.e., their errors are less “wrong”).”

6) I would be nice, although maybe not essential, to see a proof of concept that the custom RNN model can work from raw images too. The highly-processed symbolic input seems like a step back from where neural network modeling is today. Thus, I suggest a supplemental simulation where the model processes the raw image stimuli, e.g., through a pre-trained image encoder.

Thank you for this suggestion. We now discuss/motivate our design decision and mention visual stimuli as avenue for future research (Discussion, L353-362):

“Furthermore, we have chosen to work with an input representation that we deemed easiest to process for each type of learning system. Since the particular way in which agents represent the visual world was not the object of the current study, our rationale here was to provide each learning system with a representation that is easiest or most natural to process. Human participants would likely have had a harder time finding patterns in attribute-value vectors consisting of 6 numbers, than in short video clips on moving objects. In contrast, operating on raw pixels is expected to introduce more difficulty for machine learning models in terms of disentangling representations (Lazaridou et al. 2018). Future work could examine whether neural nets segment visual stimuli in a similar way as humans in grounded language learning.”

7) "Semantic difference is calculated as described above under Input Languages" (pg. 16). I couldn't find a description of this calculation, and instead went back to ref. 27.

Thank you for pointing this out. We have adjusted the pointer to the correct section heading 'Topographic Similarity to Quantify Compositional Structure'. Moreover, we have included Semantic Difference in the newly added Glossary Table for added clarity.

NCOMMS-24-13508A Response to Reviewers

Dear Editor,

Please find our response to the reviewers in blue below.

Best wishes,
Lukas Galke

REVIEWERS' COMMENTS

Reviewer #1 (Remarks to the Author):

Thank you to the authors for this detailed rebuttal. The proposed changes address all of our substantial concerns, and make a more clear and compelling case for the authors' interpretation of these modeling results.

We have only one more minor recommendation, which is that the discussion on compositionality and variation would be strengthened by considering or at least referring to this relevant recent ICLR citation:

<https://openreview.net/pdf?id=-Yzz6vIX7V-> (Conklin and Smith, Compositionality with Variation Reliably Emerges in Neural Networks, ICLR 2023).

We would also recommend a final review for consistent formatting (e.g. opening and closing quotation marks, single and double quotes) and typos.

We have added the requested citation and fixed the formatting of quotation marks.

Aside from these minor points, we look forward to the final publication of this submission.

Reviewer #1 (Remarks on code availability):

The code is appropriately documented.

Reviewer #3 (Remarks to the Author):

Reviewer #4 (Remarks to the Author):

I am satisfied with the revisions and responses to my comments. I would be happy to see the article accepted.